# ECOD: Classification of domains in AFDB Swiss-Prot structure predictions

**R. Dustin Schaeffer**[1*], **Jing Zhang**[2,3], **Qian Cong**[1,2,3], **Nick V. Grishin**[1,4]

1 Department of Biophysics, University of Texas Southwestern Medical Center, Dallas, Texas, United States of America, 2 Eugene McDermott Center for Human Growth and Development, University of Texas Southwestern Medical Center, Dallas, Texas, United States of America, 3 Harold C. Simmons Comprehensive Cancer Center, University of Texas Southwestern Medical Center, Dallas, Texas, United States of America, 4 Department of Biochemistry, University of Texas Southwestern Medical Center, Dallas, Texas, United States of America

* Richard.Schaeffer@UTsouthwestern.edu

## Abstract

The development of highly accurate protein structure prediction algorithms has led to an explosion of structural data, transforming our understanding of protein structure-function relationships across diverse organisms. Domain classifications such as the Evolutionary Classification of Protein Domains (ECOD) have incorporated these computational predictions alongside experimental structures to create comprehensive resources for the research community. The AlphaFold Protein Structure Database (AFDB) plays a unique role, providing millions of predicted structures that ECOD has systematically classified for human proteins, small pathogens, and reference proteomes. Here, we extend this classification framework to the UniProtKB/Swiss-Prot dataset, applying the Domain Parser for AlphaFold Models (DPAM) pipeline to classify domains from over 542,000 Swiss-Prot protein structure predictions, resulting in more than 1,032,000 classified domains. These domains span 3,493 ECOD topologies and display high assignment confidence (mean DPAM probability: 0.992), with extensive taxonomic and functional diversity. Notably, over 100,000 domains lack existing Pfam mappings, reflecting the extended sensitivity of structure-based classification and identifying domain groups not yet captured by sequence-based profiles. These results significantly expand ECOD's coverage into a functionally and taxonomically diverse protein space, anchoring high-confidence structure predictions in an evolutionary framework. By integrating Swiss-Prot predictions, we enhance the utility and interpretability of AlphaFold models and establish a foundation for future large-scale, functionally informed domain classifications.

**Data availability statement:** All domain classification data, figure generation scripts, and supporting CSV inputs used in this study are available at Zenodo: https://doi.org/10.5281/zenodo.16856663. This includes the full set of domains parsed from AlphaFold-predicted SwissProt proteins using DPAM, the subset of domains accessioned into ECOD (http://prodata.swmed.edu/ecod), and all data files required to reproduce manuscript figures. Accessioned domains are also publicly available as part of the ECOD database where they are integrated into the broader ECOD hierarchy.

**Funding:** This work was supported by grants from the National Institute of General Medical Sciences (GM147367 to R.D.S.), the National Institute of Allergy and Infectious Diseases (1K99AI180984-01A1 to J.Z.), the National Science Foundation (DBI 2224128 to N.V.G.), and the Welch Foundation (I-1505 to N.V.G.; I-2095-20220331 to Q.C.). Computational resources were provided by NSF ACCESS (allocations BIO250039 to Q.C. and MED240004 to N.V.G.) and TACC Lonestar6 (allocations MCB24018 and MCB23014 to N.V.G.). The funders had no role in study design, data collection and analysis, decision to publish, or preparation of the manuscript.

**Competing interests:** The authors have declared that no competing interests exist.

## Author summary

Large-scale protein structure prediction using deep learning has revolutionized our ability to study protein families and infer biological function. However, connecting these predicted structures to well-understood evolutionary classifications remains challenging. In this work, we apply a domain parsing pipeline to classify over half a million AlphaFold-predicted Swiss-Prot proteins into evolutionary groups using ECOD, a structure-based domain classification system. This enables the systematic integration of structural predictions with functional annotation across a broad range of species. Our analysis reveals taxonomic and functional diversity, highlights domain clusters with no prior annotation, and expands ECOD coverage with high-confidence, evolutionarily meaningful predictions.

## Introduction

Protein domains are independent evolutionary units, identified from either sequence, experimental structures, or structural predictions [1–3]. Studying domains and their homology is a powerful tool for understanding protein function [4–6]. Homologous domains can share function, and the propagation of functional annotation from experimentally characterized proteins and their domains to their homologous yet hypothetical or uncharacterized domains can lead to biological insights [7,8]. Protein domain classifications determine and organize these homologous domains and either fall into sequence classifications such as Pfam [9], CDD [10], or SUPERFAMILY [11] that partition protein sequence into domains and derive their taxonomy principally by sequence similarity measures or structure classifications such as SCOP [12], CATH [13], or ECOD [14] that use structural similarity to determine more distant homology (at the cost of access to fewer proteins).

ECOD organizes protein domains into a hierarchical classification with five levels. At the broadest level, Architecture groups (A-groups) describe the overall secondary structure composition of a domain (e.g., mostly alpha, mostly beta, alpha+beta). X-groups represent possible homology groups – domains that share structural features suggestive of common ancestry but without definitive evidence. H-groups (homology groups) contain domains with clear evolutionary relationships established through sequence, structural, or functional evidence. T-groups (topology groups) further subdivide H-groups by specific structural topology. Finally, F-groups (family groups) correspond to sequence families and are determined by the Pfam classification: each F-group maps to a single Pfam family or a composite of non-overlapping Pfam families, making this level dependent on external curation by the Pfam consortium rather than ECOD internal criteria. Domains that match to an ECOD topology but lack a corresponding Pfam family are assigned at the T-group level only (and receive a '.0' pseudo group F-id), awaiting future family definition. This hierarchical organization allows ECOD to capture evolutionary relationships at multiple levels of divergence, from recent sequence similarity (F-group) to ancient structural homology (X-group).

The advent of highly accurate structure prediction algorithms has led to an explosion of structural data, transforming our ability to understand protein structure-function relationships across diverse organisms. The outstanding performance of AlphaFold2 at CASP14 [15] and the subsequent development of software such as RoseTTAFold [16], ESMFold [17], and AlphaFold3 [18] has led to the widespread prediction of large protein sets resulting in resources such as the AlphaFold Protein Structure Database (AFDB) [19,20] and ESM-Atlas [17]. Domain classifications such as ECOD have incorporated these computational predictions alongside experimental structures to create comprehensive resources for the research community. The AFDB is a valuable tool in structural biology, providing millions of predicted structures that ECOD has systematically classified for human proteins [21], small pathogens [22,23], and reference proteomes [24]. AFDB has also published a set of 200M protein predictions covering the known protein space of cellular organisms [25], and this set has been classified into domains and partially incorporated into the CATH domain classification [13].

UniProtKB/Swiss-Prot represents a unique intersection of functional, taxonomic, and now structural information [26]. Integrating AlphaFold structure predictions into UniProt allows us to directly incorporate its functional annotation into our structural classification. While resources like InterPro already excel at integrating numerous domain classifications - Pfam [9], Gene3D [27], PROSITE [28], CDD [10,29] - and include some structural resources, they primarily rely on sequence-based approaches.

Swiss-Prot entries contain comprehensive functional annotations through manual curation [26], whereas UniProt entries from reference proteomes include a mixture of curated and automatically annotated sequences. The types of functional annotations (such as EC [30] and GO [31,32]) in which UniProtKB is diverse can enrich downstream bioinformatics resources such as domain classifications, potentially revealing structure-function relationships.

Here, we extend this classification framework to the UniProtKB/Swiss-Prot dataset, a manually curated collection of protein sequences spanning the tree of life. We use the Domain Parser for AlphaFold Models (DPAM) pipeline [33] to classify domains from more than 542,000 Swiss-Prot protein structure predictions, resulting in over 1,032,000 classified domains. Taxonomic diversity analysis reveals significant variation in domain distribution patterns across major lineages, with domains spanning 3 superkingdoms, 91 phyla, and 10,254 species. We classify domains from these proteins into 3,493 ECOD topology groups, 88.3% of the total classification. Our domain predictions show high assignment confidence levels, with 84.1% of domains having DPAM probabilities greater than 0.9 (mean probability: 0.992). Clustering at 40% sequence identity yields 158,942 domain clusters with an average of 6.5 members per cluster. Notably, over 100,000 domains lack existing Pfam mappings. Analysis of these domains reveals that this gap primarily reflects differences in detection sensitivity between structure- and sequence-based methods, with structural classification identifying domain sub-groups that precede their formalization as sequence families. In the following sections, we demonstrate that AlphaFold-predicted structures of Swiss-Prot proteins display exceptionally high confidence scores (median pLDDT 91.35), enabling reliable domain annotation across functionally and taxonomically diverse predicted protein structures. We analyze how these newly classified predicted domains complement existing pools of experimental domains and those from reference proteomes, and how this clustering approach allows us to stratify predictions by confidence and identify areas of protein space with strong experimental support versus those that would benefit from additional experimental validation, providing a more comprehensive structural perspective on protein diversity across the tree of life.

## Results and discussion

### Properties of AFDB Swiss-Prot predictions

The ECOD 48 proteomes set (48P) and Swiss-Prot show similar overall size (~564,449 vs. 542,378 proteins) and length distribution patterns, with both datasets spanning identical extremes (16–2699 residues). However, 48P exhibits longer sequences (median 335 residues) compared to Swiss-Prot (median 295 residues) (Fig 1A). An anomaly appears around 1500 residues in the 48P dataset (due to windowed predictions for very large proteins). These differences reflect their distinct curation approaches: while Swiss-Prot prioritizes well-characterized proteins across diverse organisms, the 48P

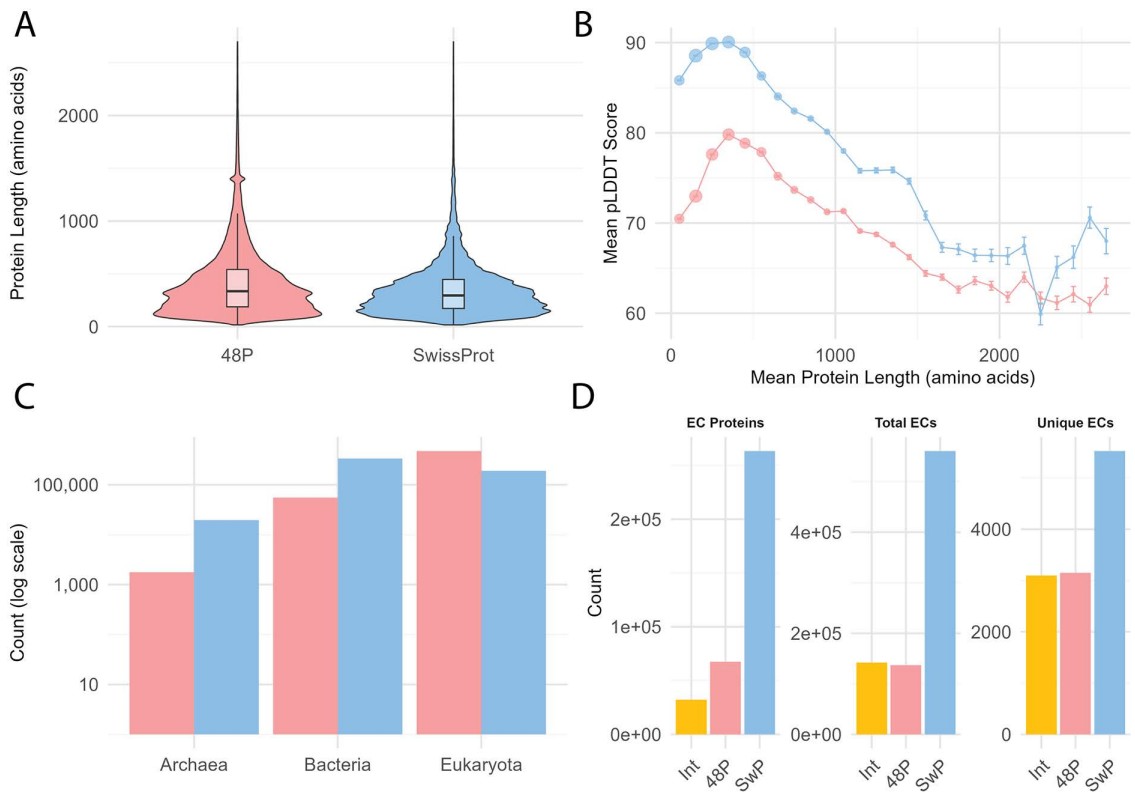

**Fig 1. Comparison of physical, computational, taxonomic, and functional properties between the Swiss-Prot and 48P datasets. A)** Protein length distributions showing the 48P dataset's longer sequences (median 335 residues) compared to Swiss-Prot (median 295 residues). **B)** Relationship between protein length and pLDDT scores, illustrating that AlphaFold prediction confidence is higher in Swiss-Prot across most length ranges. **C)** Taxonomic distribution across superkingdoms, with Swiss-Prot showing broader phylogenetic coverage than 48P. **D)** Distribution of enzyme annotations by dataset, comparing proteins with at least one EC number, total EC annotations, and unique EC identifiers observed.

incorporates proteins that tend to be longer or contain more disordered regions. The overlap between these sets primarily consists of highly curated model organisms, with Swiss-Prot's non-overlapping proteins providing additional functional, enzymatic, and taxonomic insights from related organisms.

This length distribution difference directly impacts prediction confidence, with Swiss-Prot showing substantially higher AlphaFold pLDDT scores (median 91.35) than 48P (median 78.01) across all confidence bands (Fig 1B). Most Swiss-Prot proteins (56.82%) fall in the very high confidence range (≥90), compared to only 17.76% of 48P proteins. Both datasets exhibit a complex relationship between protein length and prediction confidence, with an inverted U-shaped pattern peaking at moderate lengths (250–400 amino acids) before declining for longer proteins. Despite following similar trends, Swiss-Prot maintains 8–10 points higher pLDDT scores across all length ranges.

The taxonomic distribution further illustrates the complementary nature of these datasets (Fig 1C). Swiss-Prot covers proteins from 10,854 species across 98 phyla, with a majority of bacterial (62%) and eukaryotic (35%) proteins, while archaeal proteins make up the remainder. In contrast, 48P focuses heavily on eukaryotes (89%) with lower bacterial (10%) representation. The per-organism coverage necessarily differs, with Swiss-Prot providing an average of 50 proteins per organism (median: 2) and 48P offering deep coverage of 11,227 proteins per organism (median: 8,652).

Swiss-Prot demonstrates broader enzymatic coverage with 5,520 unique EC numbers compared to 3,148 in 48P, though they share a substantial core of 3,096 EC numbers (Fig 1D). Swiss-Prot has nearly four times more

enzyme-annotated proteins (263,138 vs. 67,361) and higher annotation completeness (85% vs. 80.29% complete EC numbers). While both datasets show similar distributions across enzyme classes, with Transferases (EC 2) and Hydrolases (EC 3) dominating, Swiss-Prot maintains more balanced representation across all seven enzyme classes. This taxonomic and functional complementarity enhances the value of combining these datasets for comprehensive domain classification.

## DPAM domain classification of AFDB Swiss-Prot proteins

Our analysis identified more than 1,032,000 classified domains from over 542,000 Swiss-Prot protein structure predictions. These domains exhibit a balanced distribution between single-domain proteins (50.1%) and multi-domain proteins (49.9%), with an average of 1.96 domains per protein (Fig 2A). The distribution declines sharply as domain count increases, with two-domain proteins representing 28.6% of the dataset, three-domain proteins 11.3%, and progressively smaller proportions for higher domain counts. The maximum observed domain count was 42, found in Notch proteins across various vertebrate species.

Our analysis also identified 15,382 proteins without domain assignments, representing approximately 2.8% of the Swiss-Prot dataset. These unclassified proteins exhibit a strong skew toward shorter sequences, with a median length

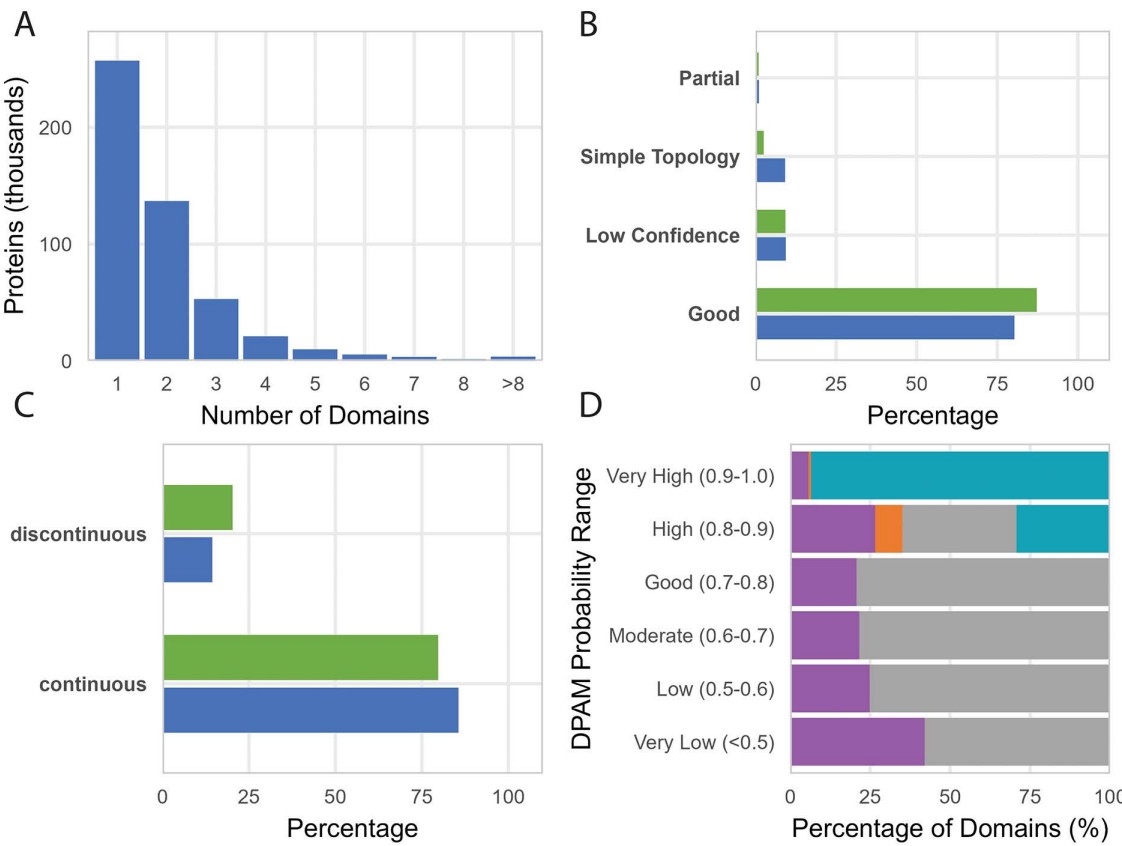

**Fig 2. Domain density of DPAM classifications of AFDB Swiss-Prot. A)** Distribution of the number of domains per Swiss-Prot protein. **B)** Distribution of dom0061in classification categories—Good, Partial, Simple Topology, and Low Confidence—shown by domain count (blue) and residue count (green). **C)** Proportion of domains that are continuous or discontinuous in sequence space, measured by count and by residues. **D)** Distribution of domain categories across DPAM probability ranges, colored by judge category (green: good_domain, yellow: low_confidence, orange: simple_topology, red:partial_domain) Most high-confidence assignments fall into the Good category by design.

of 84 residues and 80% being under 200 residues. A list of these Swiss-Prot proteins lacking domains is distributed as a benchmark set with our domain classification. We suspect they are a combination of *bona fide* disordered regions lacking domains, possible novel domains missed by DPAM, prediction errors, and/or genome annotation errors.

DPAM domains are categorized post-assignment based on their DPAM confidence, secondary structure content, and alignment coverage of the reference domain. The classification quality metrics demonstrate the robustness of our approach, with 80.34% of domains categorized as well-assigned (i.e., "good_domain") with high DPAM confidence, fractional secondary structure element (SSE) content, and good alignment coverage to their reference hit (Fig 2B). When measured by residue count rather than domain count, well-assigned domains represent 87.25% of all residues, suggesting these tend to be larger domains. Simple topology domains (9.20% of domains but only 2.54% of residues) typically represent smaller structural units, while partial domains with high assignment confidence but low alignment coverage represent fragments or incomplete structural units. The relationship between DPAM probability and judge category is shown in Fig 2D: well-assigned domains dominate at high DPAM probabilities (> 0.85), while low-confidence and simple topology domains are distributed across lower probability ranges, reflecting that judge assignment is directly determined by DPAM confidence thresholds.

Domain continuity analysis reveals that 85.60% of domains are continuous in sequence space, comprising 79.70% of residues (Fig 2C). The remaining 14.40% are discontinuous domains, accounting for 20.30% of residues, reflecting domains that incorporate multiple sequence segments into a single structural folding unit. This disproportionate residue percentage indicates discontinuous domains tend to be larger on average, and their identification is critical for accurately classifying evolutionarily related proteins that have undergone insertions or fusions.

Taxonomic distribution across major homologous groups reveals evolutionary patterns and specialization trends (Fig 3A). While ancient, functionally essential architectures like P-loop domains and Rossmann folds maintain a significant presence across all superkingdoms, they skew heavily toward bacteria (70–74%). Notable exceptions include ARM repeats, which show overwhelming eukaryotic specificity (91%), reflecting their specialized roles in nuclear transport and signaling pathways unique to nucleated cells. Similarly, immunoglobulin-related domains demonstrate strong eukaryotic

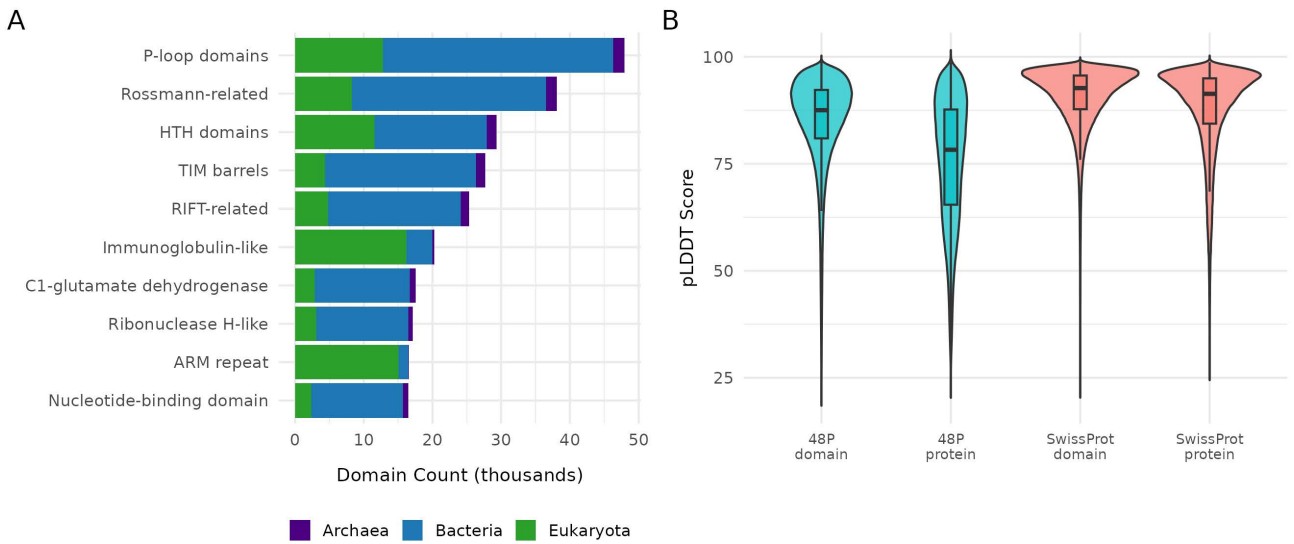

**Fig 3. Taxonomic distribution of protein domain classifications from the Swiss-Prot dataset. A)** Distribution of the top 10 most abundant ECOD H-groups across superkingdoms. Expected eukaryotic enrichments (e.g., immunoglobulin-like domains, ARM repeats) are observed alongside bacterial-dominant folds. **B)** pLDDT score distributions for full proteins and parsed domains in both 48P and Swiss-Prot datasets. Domain parsing regularizes prediction confidence and reduces length-associated variability.

preference (80%), aligning with their functions in complex immune systems. Helix-turn-helix (HTH) domains present a more balanced distribution, highlighting their fundamental importance as DNA-binding motifs across diverse organisms.

Among proteins containing duplicated domains, several superfamilies stand out: P-loop containing nucleoside triphosphate hydrolases (found in 5,708 proteins with an average of 2.09 duplications), ARM repeats (3,705 proteins, average 2.75 duplications), and zinc finger domains (1,985 proteins with an average of 6.84 duplications, maximum 38 duplications). This pattern of domain duplication is a common evolutionary strategy independently utilized across diverse protein families.

The pLDDT confidence distributions compare protein versus domain confidence scores across Swiss-Prot and 48P datasets (Fig 3B), demonstrating that domains generally show higher and more consistent confidence scores than whole proteins. This "smoothing effect" validates our domain parsing approach by confirming it successfully identifies more consistently structured regions within structure predictions of proteins. Classification confidence varies across phyla, with some lineages showing unexpected patterns that may reflect biological variation or methodological artifacts (S1 Fig). To validate these taxonomic patterns before incorporating domains into ECOD, we clustered domains by sequence similarity and mapped them to Pfam families, serving both validation and discovery functions.

## Sequence-based clustering of DPAM domains

We clustered the DPAM domains at multiple sequence thresholds [40%, 70%, 99%] for several reasons: 1) to investigate the overall sequence redundancy within the set, particularly within highly duplicated domains such as zinc fingers, EGF domains, and notch repeats; 2) to more carefully screen for potential domain boundary issues by examining clusters with high sequence redundancy but inconsistent multiple sequence alignment; and 3) to analyze the diversity of domain assignment, both through external sequence classification to Pfam and internal taxonomic diversity.

Our clustering approach included domains from all DPAM judge categories, allowing us to examine clustering behavior across confidence levels. Interestingly, our analysis revealed that domains from all judge categories clustered efficiently at all sequence identity thresholds, suggesting that even domains with lower classification confidence exhibit recognizable sequence relationships.

At 99% sequence identity, the clustering produced 758,709 clusters, with 85.79% of these being singletons (Fig 4). As the threshold decreased to 70%, the number of clusters reduced by approximately half to 380,622, with singletons representing 63.21% of clusters. At 40% identity, further consolidation occurred, resulting in 158,942 clusters with singletons comprising 43.47%. Correspondingly, the maximum cluster size increased dramatically from 147 domains at 99% identity to 3,382 domains at 40% identity, demonstrating how lowering the sequence identity threshold reveals more distant evolutionary relationships. The data reveal a clear relationship between sequence identity thresholds and domain clustering patterns. Importantly, while singletons comprise 43.5% of clusters at 40% identity, they represent only 6.7% of all domains (69,908 of 1,032,610); the remaining 93.3% of domains successfully cluster with at least one other domain. Among well-assigned domains, 97.9% of singletons at 40% are also singletons at 70%, indicating that these are genuine unique sequences rather than borderline threshold cases.

Our taxonomic analysis of these clusters showed a clear relationship between sequence identity threshold and taxonomic diversity. At 99% identity, nearly all clusters contained domains from a single phylum, consistent with recent evolutionary divergence (Fig 5). As the threshold decreased to 40%, approximately 40% of clusters spanned multiple phyla, indicating the capture of more ancient evolutionary relationships. This pattern was quantified using a weighted taxonomic diversity score (see Methods) scaled from 0-1; the fraction of clusters with diversity >0.7 increased substantially at lower sequence identity thresholds.

Family groups were generated against Pfam (see Methods) using HMMER. Classification against Pfam at this stage provides a degree of external validation (some Pfam families have been generated and informed by previous ECOD outputs). We expect that many Swiss-Prot proteins and domains, being the target of previous extensive classification, will

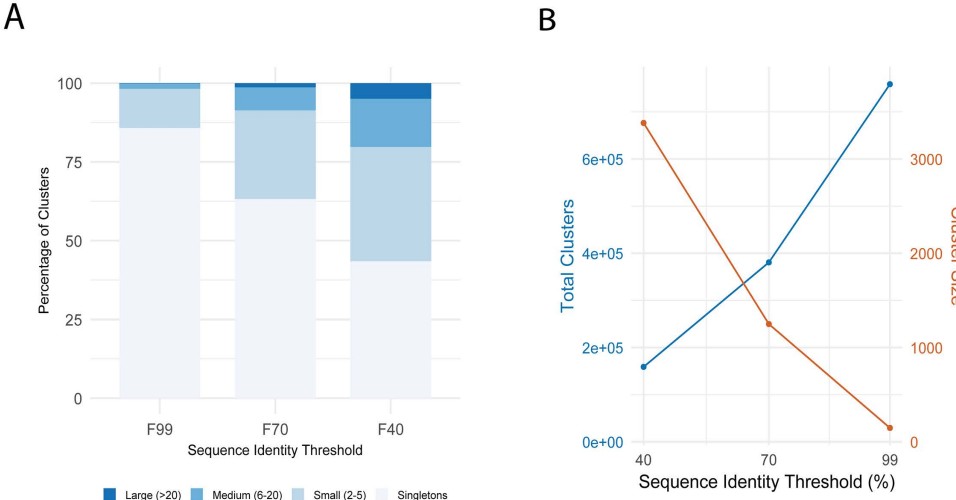

**Fig 4. Sequence clustering of Swiss-Prot DPAM domains at 40%, 70%, and 99% sequence identity clustering using CD-HIT. A)** Distribution of cluster sizes at 99%, 70%, and 40% sequence identity. Most clusters at 99% identity are singletons, while lower thresholds result in more consolidated, larger clusters. **B)** Total number of clusters (blue, left axis) and maximum cluster size (orange, right axis) as a function of sequence identity threshold. Lower thresholds capture broader homology relationships.

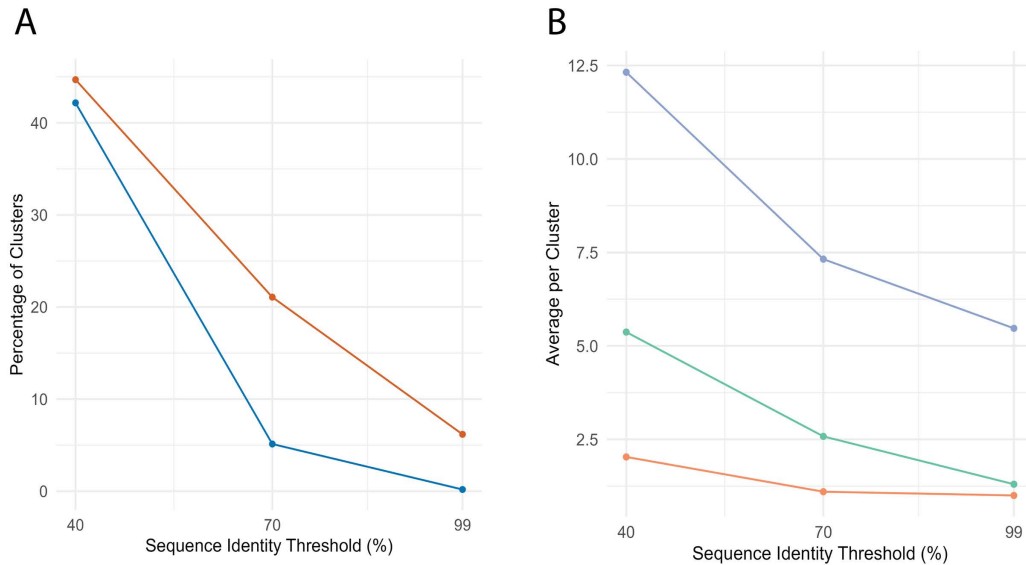

**Fig 5. Taxonomic breadth of domain sequence clusters. A)** Percentage of clusters at each identity threshold containing high taxonomic diversity (orange) or spanning multiple phyla (blue). **B)** Average number of species (blue), families (cyan), and phyla (orange) per cluster across thresholds. Clusters become increasingly taxonomically broad as identity threshold decreases.

map well into known sequence families. We also hope that there will be gaps in classification that the addition of structural domain predictions may aid. Where domains may be assigned with low confidence by DPAM to ECOD, they might have confident classification to Pfam, giving us external support to classify or manually curate ECOD. Where we have confident classification to ECOD but no classification to Pfam, we can forward these domains to Pfam, where they may be used

as the basis for new sequence families. Although we mapped all DPAM classes in this pass, only well-assigned DPAM domains were accessioned into ECOD at this time.

Well-assigned DPAM domains map exceptionally well to Pfam (93.6%), demonstrating strong concordance between structural and sequence-based classification for high-confidence domains (Fig 6A). Simple topology domains have many opportunities to be added to Pfam (59.5% unmapped) but also reveal divergence between sequence- and structure-based classifications in short domains, where non-secondary-structure elements such as metal-binding sites or disulfide bonds may define structure more strongly than secondary structure content. Interestingly, most unmapped domains have high prediction confidence (pLDDT > 80), countering the assumption that unmapped regions are primarily disordered or poorly predicted (Fig 6B).

To understand the sources of the Pfam mapping gap among the 52,644 unmapped well-assigned domains, we examined whether these domains occupy regions of structural classification space that Pfam covers elsewhere. We find that 98.9% of unmapped well-assigned domains belong to ECOD T-groups where other domains successfully map to Pfam, and only 1% reside in T-groups with no Pfam representation at all. The gap therefore does not primarily reflect structural novelty, but rather the limits of sequence-based detection within known structural families.

We further classified unmapped domains by comparing their ECOD structural reference hits to those of their Pfam-mapped siblings within the same T-group. Of the 52,644 unmapped well-assigned domains, 85.7% (45,129) match the same ECOD reference domains as Pfam-mapped members of their T-group, indicating that they are sequence-divergent members of structurally characterized families where an appropriate Pfam HMM exists but cannot detect these sequences. These domains are shorter (median 105 vs. 149 residues for mapped domains), more frequently discontinuous (21.5% vs. 13.7%), and represent the detection sensitivity boundary between structural and sequence-based methods.

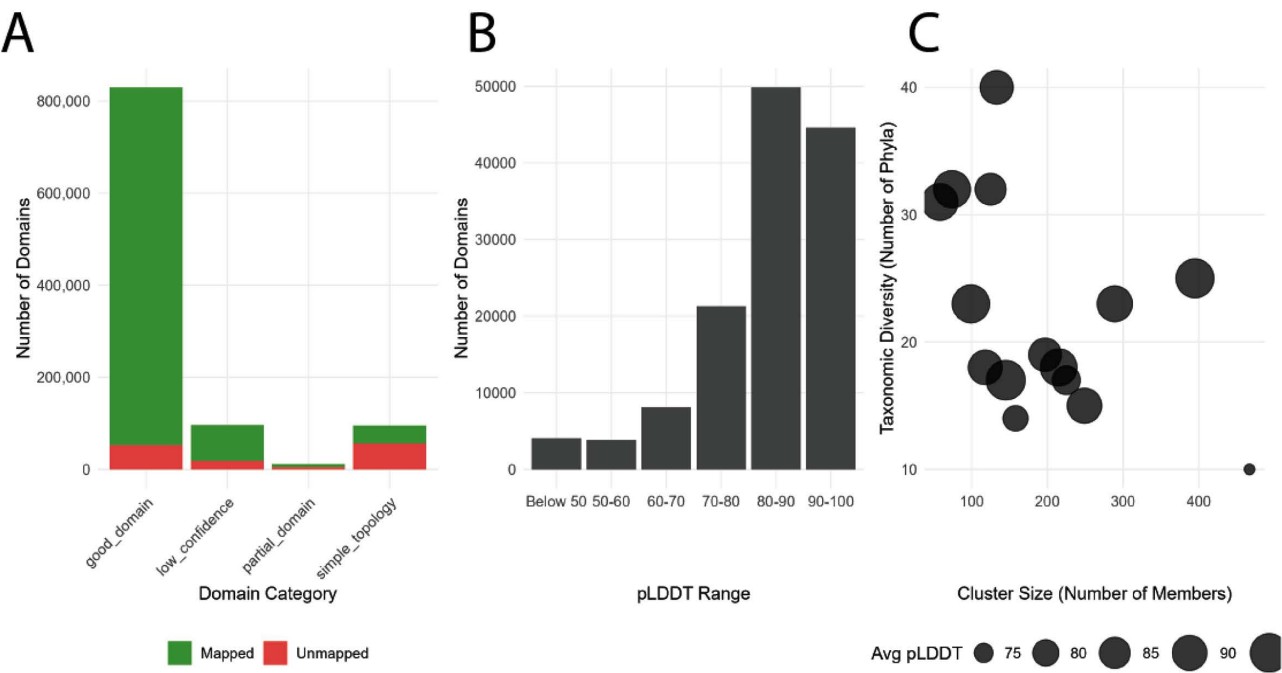

**Fig 6. Pfam mapping and properties of unmapped domains. A)** DPAM domains that can be mapped to Pfam (green) and that remain unmapped (red) **B)** Average pLDDT of DPAM domains that cannot be mapped to Pfam **C)** Taxonomic diversity, cluster size, and average pLDDT of standout sequence clusters lacking Pfam mapping, ideal candidates for definition of new Pfam families.

The remaining 14.3% (7,515) match ECOD structural references not used by any Pfam-mapped domain – structurally classified sub-groups within known T-groups that lack corresponding sequence profiles. These are not structural outliers, they have high classification confidence (median HH probability 1.0) and are well-bounded within existing ECOD topology groups. Rather, they represent domain groups where structural classification has resolved evolutionary relationships that sequence-based methods have not yet formalized.

To test whether this gap is closing, we scanned these unmapped sequences against Pfam releases from version 37.4 through 38.2. This includes Pfam releases that followed our analysis and subsequent to our communication of these unmapped domains to the Pfam curators. Pfam 38.2 (30,134 families) captures 16.7% (8,797) of these previously unmapped domains, with 91% of these newly captured domains (7,990) attributed to 2,090 families created after our original analysis. Several of these new families correspond directly to the structural sub-groups identified above: CPSase_L_D1 (PF25596, 468 domains) fills a gap in the PreATP-grasp T-group, AAA_lid_14 (PF25601, 127 domains) in the AAA+ATPase lid T-group, and HisZ_C (PF27460, 94 domains) in the Class II aminoacyl-tRNA synthetase anticodon-binding T-group. Domains in underspecified sub-groups are captured at a higher rate (23.4%) than those in the sensitivity-gap category (15.6%), consistent with Pfam prioritizing the creation of new families for structurally coherent groups.

Clustering analysis revealed several large unmapped domain clusters with over 100 members and broad taxonomic distributions (15 + phyla). A particularly diverse cluster (133 members from 40 phyla) corresponds to an intermediate domain (ECOD T: 593.1.1) from GroEL-like chaperone proteins found across eukaryotes, bacteria, and archaea (Fig 6C). Although the Pfam model Cpn60_TCP1 (PF00118) captures many of these proteins as a single unit, it does not resolve the finer-grained structural domains defined in ECOD. This exemplifies the boundary-definition component of the mapping gap, these domains are not absent due to novelty, but reflect differences in how structure- and sequence-based methods delineate multi-domain architectures (Fig 7A-7E).

Currently, our Pfam mapping analysis is primarily focused on well-assigned DPAM domains, with ongoing work to extend this to other judge categories. Domains found to be internally consistent, representative of diverse external groups, or singletons but experimentally supported were considered candidates for ECOD accession. Domains with complicated relationships or inconsistent behavior due to domain boundary issues were retained for reclassification or manual review. Altogether, 890,055 distinct DPAM domains received a Pfam mapping, 766,132 of which were well-assigned domains. These Pfam-mapping well-assigned DPAM domains mapped to 12,019 distinct Pfam sequence families. These sequence families were used to generate F-groups and F-group mappings for putative domains. Domains mapped to existing F-groups in ECOD where possible, and assigned to newly generated F-groups where a Pfam family or composite of Pfam families had not previously been observed under that H/T-group. 260 new ECOD F-groups were created (233 simple families and 27 composite families) to hold these domains with a single high-quality DPAM domain promoted to serve as representative for each group. An additional 83,205 domains were assigned at the T-group level only, representing candidates for future F-group creation.

## Candidate domains for ECOD accession

We aim to recruit the highest quality domains from structural predictions to combine with our existing experimental classification of domains. Here, we initially automatically populated ECOD with domains from the Swiss-Prot DPAM set with the highest DPAM assignment confidence. As there is overlap between the AFDB Swiss-Prot and AFDB 48 Proteomes set, we favored existing domains over new, preventing new overlapping domains from being accepted into ECOD. 168,204 domains from the Swiss-Prot set were deferred from ECOD entry based on existing AFDB protein classification in ECOD. Of these deferred domains, 97% had identical T-group classifications to their existing counterparts and 79% had identical ranges. Excluding those overlapping domains identified above, our analysis identified 648,979 candidate domains (approximately 78.2% of all well-assigned domains in the Swiss-Prot set) from 369,639 unique proteins (approximately 70.1% of all proteins in the Swiss-Prot set). These candidate domains represent the non-overlapping set of Swiss-Prot

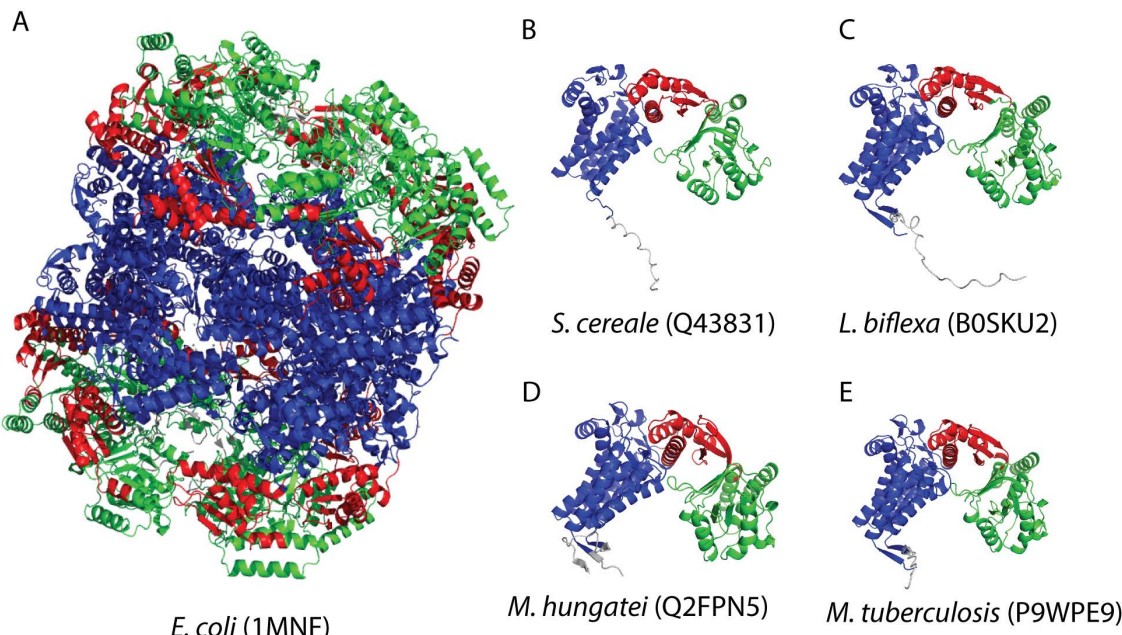

**Fig 7. DPAM domains homologous to GroEL/Cpn60 lacking Pfam mapping. A)** Experimental structure of the *E. coli* GroEL complex (PDB: 1MNF), with equatorial (blue), intermediate (red), and apical (green) domains shown across subunits. Predicted structures of taxonomically diverse GroEL homologs from **B)** *S. cereale* (Q43831), **C)** *L. biflexa* (B0SKU2), **D)** *M. hungatei* (Q2FPN5), and **E)** *M. tuberculosis* (P9WPE9), each displaying conserved architecture despite lacking Pfam mappings. These examples highlight the complementarity of predicted and experimental structures and demonstrate that unmapped domains may correspond to well-characterized folds.

predictions not previously classified and were targeted for accession into ECOD. As expected, common model organisms classified previously in our 48 proteomes set have few additional domains. For example, human proteins have only 481 candidate domains out of 51,757 total domains (0.13%), and mouse with 423 candidates from 42,390 domains (0.18%). We found more novelty in less-studied bacterial strains, with some having over 90% of their domains as candidates - for example, *Photobacterium profundum* has 1,053 candidate domains out of 1,110 total domains (94%), and *Vibrio vulnificus* has 2,547 candidates from 2,708 domains (94%). These novelties fill gaps in the taxonomic distribution of the ECOD database. Of these candidate domains, 619,944 (95.5%) could be successfully mapped to Pfam families, providing a solid foundation for classification. The remaining 29,035 domains (4.5%) lack Pfam mappings. As detailed above, these unmapped domains are predominantly sequence-divergent members of structurally characterized families (85.7%) or members of structural sub-groups not yet formalized as Pfam sequence families (14.3%), rather than structurally novel entities. The progressive capture of these domains by recent Pfam releases (16.7% by v38.2) demonstrates active convergence between structural and sequence-based classifications, with ECOD's structural assignments effectively guiding the creation of new sequence families.

The Swiss-Prot DPAM pipeline classifies domains into the existing ECOD hierarchy; no new architectures or X-groups are created through this automated process. All 20 ECOD architectures were expanded, with the largest contributions to the alpha/beta three-layered sandwiches (S1 Table). At the X-group level, P-loop domains (ECOD X: 2004, +34,856 domains), Rossmann-like domains (ECOD X: 2003, +31,705) and TIM barrels (ECOD X: 2002, +22,673 domains) showed the greatest expansion (S2 Table). Taxonomically, bacterial proteomes dominated (76.9% of domains added), with most X-groups showing 70–90% bacterial composition. Notable exceptions include the Immunoglobulin-like beta-sandwiches (ECOD X: 11, 60.7% eukaryotic) and the repetitive alpha-hairpins (ECOD X: 109, 83.8% eukaryotic), reflecting the

prevalence of these folds in metazoan immune systems and repeat-containing proteins (S3 Table). ECOD v293 includes 260 new F-groups, of which 110 (42%) contain Swiss-Prot domains, spanning 50 distinct parent T-groups (S4 Table). A total of 29,720 domains (4.6%) remain classified at the T-group level (*.0 groups in ECOD) awaiting F-group assignment, with the largest accumulations in structurally diverse superfamilies such as Immunoglobulin/Fn3/E-set (1,002 domains) and ARM repeats (843 domains) (S5 Table).

These results represent a major expansion of ECOD's structural coverage, particularly in taxonomically diverse and previously under-characterized regions of protein space. By anchoring AlphaFold predictions from Swiss-Prot in a curated evolutionary framework, this work establishes a foundation for future ECOD updates. In particular, the unmapped high-confidence domains identified here – spanning 2,090 structural sub-groups subsequently formalized as Pfam families in version 38.2 alone- demonstrate that structure-based classification systematically identifies domain groups ahead of sequence-based methods. These results establish a framework for prioritizing the creation of new sequence families and guide the continued integration of structural and functional annotations across ECOD and Pfam.

## Methods

### DPAM classification of Swiss-Prot proteins

We downloaded a tarball of gzipped mmCIF AlphaFold predictions from the AFDB. We acquired prediction-aligned errors (PAE) files separately by scripted individual downloads. Domains were classified using the DPAM pipeline described elsewhere [33]. Briefly, proteins are partitioned into putative domains using a combination of interresidue properties, after which putative domains are assigned to the ECOD reference using a neural network combination of sequence and structural alignment results. Following their assignment, DPAM domains receive a categorical assessment based on DPAM confidence, DSSP [34,35] secondary structure content, and alignment coverage. Domains are classified as 'good_domain' (high-confidence match, ≥ 3 SSEs), 'partial_domain' (high-confidence but incomplete coverage, ≥ 3 SSEs), 'low_confidence' (uncertain assignment, ≥ 3 SSEs), or 'simple_topology' (<3 SSEs).

Following initial data generation, DPAM intermediate files were loaded to a PostgreSQL 13.3 database for exploratory data analysis. R/RStudio was used for data analysis in concert with Claude (see below). Protein structures were generated using PyMOL. Domains were incorporated into the main ECOD classification where they did not conflict with existing classifications. Specific classification outputs for the AFDB Swiss-Prot set are provided in the Zenodo repository associated with this manuscript.

### Sequence clustering of DPAM domains using CD-HIT

DPAM domains were clustered by sequence using CD-HIT [36–38] at 40%, 70%, and 99% thresholds using a bandwidth of 20, min_length of 10, and tolerance of 2. Word lengths varied over sequence thresholds: 5, 4, and 2 for F99, F70, and F40, respectively. Cluster members and cluster FA files are deposited in the Zenodo repository associated with this set.

Taxonomic diversity of domain clusters was assessed using a weighted multi-level approach. Domains were clustered at three sequence identity thresholds (99%, 70%, and 40%), referred to as F99, F70, and F40, respectively. For each cluster, taxonomic diversity was quantified using a composite score calculated by: 1) Determining the distribution of unique taxa at distinct taxonomic levels (Species - 30%, Family - 25%, Order - 20%, Class - 15%, and Phylum - 10%). 2) For each taxonomic level, calculate the ratio of distinct taxa to the total number of members in the cluster. 3) Compute a weighted sum of these ratios to generate a final diversity score ranging from 0 to 1.

Clusters with diversity scores exceeding 0.7 were classified as "highly diverse" and likely represent domains conserved across broad taxonomic groups. Additional metrics included the percentage of clusters spanning multiple phyla, the average number of distinct taxa per cluster, and the maximum taxonomic range.

Analysis was restricted to clusters containing at least three domain members to ensure reliable diversity assessment. The taxonomic classification was based on NCBI Taxonomy, with each protein domain linked to taxonomy via its source

organism. This approach enabled quantitative comparison of domain conservation patterns across different sequence identity thresholds.

### HMMER classification of DPAM domains

DPAM domains were assigned to Pfam using the HMMER suite (hmmscan) [39]. Gathering thresholds (--cut_ga) were used to establish minimum quality hits, and the best series of non-overlapping hits above this threshold were used to assign domains to a specific sequence family. Pfam 38 was used to determine these mappings [9]. Each mapping was assigned a confidence level (high/medium/low/uncertain) based on coverage percentage and bit score thresholds, with domains exhibiting ≥80% coverage and bit scores >50 classified as high confidence. Mappings were integrated with ECOD structural classification through established Pfam-to-ECOD family correspondences where possible. This approach achieved high coverage rates across domain categories (93.65% of good_domains, 81.10% of low_confidence domains, and 40.50% of simple_topology domains), while unmapped high-quality domains with significant taxonomic distribution were flagged as candidates for novel family designation.

To characterize the sources of the Pfam mapping gap, unmapped well-assigned domains were classified by the Pfam mapping rate of their parent ECOD topology (T-group) and by whether their ECOD structural reference (hit_ecod_domain_id) was shared with Pfam-mapped domains in the same T-group. Temporal analysis was performed by scanning the 52,644 well-assigned domain sequences against Pfam-A HMM libraries from versions 28 (2015), 37.4, 38, 38.1, and 38.2 using hmmscan with gathering thresholds (--cut_ga). Families created after Pfam 38.0 (accession PF25159 and above) were classified as new families to distinguish improved detection by existing models from coverage by newly created families.

### Use of large language models

Large language model (LLM) tools, including Claude (Anthropic) and ChatGPT (OpenAI), were used to assist with manuscript preparation and analysis infrastructure. Claude was used to draft portions of the SQL schema underlying the Swiss-Prot domain analysis database and to generate initial versions of R scripts for Fig creation. Claude also contributed to evaluating manuscript structure and improving the clarity and consistency of the narrative. All outputs from these tools were reviewed, validated, and modified by the authors. All hypotheses, interpretations, and conclusions presented in this study reflect the authors' original ideas.

## Supporting information

**S1 Table. Architecture expansion from Swiss-Prot integration.** Domain counts for each ECOD architecture before and after Swiss-Prot integration, showing total expansion and percentage of domains contributed by Swiss-Prot. (XLSX)

**S2 Table. X-group expansion.** Top X-groups ranked by Swiss-Prot domain contribution, with pre- and post-integration domain counts. (XLSX)

**S3 Table. X-group taxonomic breakdown.** Swiss-Prot domain counts per X-group stratified by superkingdom (Bacteria, Eukaryota, Archaea), with percentage taxonomic composition. (XLSX)

**S4 Table. Newly created F-groups.** List of 260 F-groups created during Swiss-Prot accessioning, including domain counts, parent T-group, and Pfam identifiers for composite families. (XLSX)

**S5 Table. T-group-only domain assignments.** Domains assigned at T-group level without F-group designation, grouped by T-group with taxonomic distribution.
(XLSX)

**S1 Fig. Classification confidence by phylum.** Heatmap showing classification confidence distribution across representative phyla, grouped into confidence bins, with hierarchical clustering and annotations for superkingdom and domain count.
(TIFF)

## Acknowledgments

We thank Drs. Lisa Kinch, Kirill Medvedev, and Jimin Pei for helpful discussions.

## Author contributions

**Conceptualization:** R. Dustin Schaeffer, Jing Zhang, Qian Cong, Nick V. Grishin.

**Data curation:** R. Dustin Schaeffer, Jing Zhang, Qian Cong.

**Funding acquisition:** R. Dustin Schaeffer, Qian Cong, Nick V. Grishin.

**Investigation:** R. Dustin Schaeffer.

**Methodology:** R. Dustin Schaeffer.

**Project administration:** R. Dustin Schaeffer.

**Resources:** Jing Zhang, Qian Cong.

**Software:** R. Dustin Schaeffer, Jing Zhang, Qian Cong.

**Supervision:** Qian Cong, Nick V. Grishin.

**Validation:** R. Dustin Schaeffer.

**Visualization:** R. Dustin Schaeffer.

**Writing – original draft:** R. Dustin Schaeffer.

**Writing – review & editing:** R. Dustin Schaeffer, Nick V. Grishin.

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
