## [Decision Letter · Decision Letter 0]

14 Nov 2025

ECOD: Classification of domains in AFDB Swiss-Prot structure predictions

PLOS Computational Biology

Dear Dr. Schaeffer,

Thank you for submitting your manuscript to PLOS Computational Biology. After careful consideration, we feel that it has merit but does not fully meet PLOS Computational Biology's publication criteria as it currently stands. Therefore, we invite you to submit a revised version of the manuscript that addresses the points raised during the review process.

We look forward to receiving your revised manuscript.

Kind regards,

Chaok Seok

Academic Editor

PLOS Computational Biology

Nir Ben-Tal

Section Editor

PLOS Computational Biology

**Journal Requirements:**

3) Please amend your detailed Financial Disclosure statement. This is published with the article. It must therefore be completed in full sentences and contain the exact wording you wish to be published.

State what role the funders took in the study. If the funders had no role in your study, please state: "The funders had no role in study design, data collection and analysis, decision to publish, or preparation of the manuscript.".

**Reviewers' comments:**

Reviewer's Responses to Questions

**Comments to the Authors:**

Reviewer #1: The manuscript by Schaeffer et al., "ECOD: Classification of domains in AFDB Swiss-Prot structure predictions" applies a previously developed DPAM pipeline to classify the protein domains from the AF-predicted structures. The manuscript substantially expands the coverage of classified domains by including over 500,000 structures from Swiss-Prot.

Furthermore, the submission substantially expands the phylogenetic coverage of species for which the domains have been annotated, going far beyond the 48 model organisms previously annotated by ECOD.

Expanding ECOD with predicted structures is overall an important work, but the manuscript in the current form doesn’t fully reflect it. There are several major issues that the authors need to address before the manuscript becomes suitable for publication in PLoS Computational Biology:

1. With expanded phylogeny, ECOD should now offer an advanced phylogenetic browser/filtering, i.e., the ability to select, compare, or exclude domains across one or multiple taxonomic groups.

2. "4 superkingdoms" (line 93). The Consensus Statement by the International Committee on Taxonomy of Viruses indicates that "the current view [is] that viruses have multiple origins (polyphyly) and that their diversity cannot be represented by a single virosphere-wide tree" (doi: 10.1038/s41564-020-0709-x). Therefore, grouping viruses into a single fourth superkingdom is not supported. In line with this view, the NCBI changed the naming scheme**,** replacing "superkingdom" with "Acellular root" for viruses (https://ncbiinsights.ncbi.nlm.nih.gov/2025/02/27/new-ranks-ncbi-taxonomy/). Please comply throughout the text with the current consensus.

3. "Initial analysis shows Swiss-Prot contains a significantly higher density of functional annotations compared to reference proteomes" (lines 81-82). This should say Swiss-Prot entries and UniProt entries from reference proteomes. Additionally, is this initial analysis part of your manuscript? This is not clear; otherwise, please provide a reference.

4. The manuscript should provide a clear description of new domains identified in this study (maybe in the form of a table): which architectures and X groups expanded due to discoveries of new domains and which phylogenetic groups contributed to this expansion.

5. The authors state that "the propagation of functional annotation from experimentally characterized proteins and their domains to their homologous yet hypothetical or uncharacterized domains can lead to biological insights" (lines 54-56). Yes, but it can also lead to errors. While there is clearly value in functional annotation, please be specific and provide examples and references.

Minor comments:

6. The authors start this manuscript with the following sentence: "Protein domains are independent evolutionary units that convey function and fitness" (line 51). This appears to be an overstatement; please provide references showing that protein domains in general provide fitness or remove statements that are unspecific, vague, and unverifiable assumptions.

7. Figure 1. Panel D: labels are cropped and inconsistent. The figure legend for panel D also needs revision as the names on the plot do not correspond to the names in the legend. In the plot we have "EC Annotations", "EC Proteins", and "Unique ECs"; in the legend we have "proteins with at least one EC number", "total EC annotations", and "unique EC identifiers".

8.) Figure 2. Panel A: the y-axis title is cropped. Please fix the color codes: in Panel B, as per the figure legend, domain count is shown in blue and residue count in green. However, it is confusing that the label "Simple Topology" is shown in the same green as residue count bars, and the label "Good" is shown in the same blue as domain count bars. Are the four categories "Partial", "Simple Topology", "Low Confidence", and "Good" mutually exclusive? Please define.

9. Panel C appears to follow the same color code as panel B, but this is not specified; please clarify. The bars in panel D appear to follow the colors indicated by the labels in panel B; please clarify.

10. Please provide a definition for "Simple Topology". The text states that "Simple topology domains (9.20% of domains but only 2.54% of residues) typically represent smaller structural units." Is this a or a definition?

11. Lines 263-269 and 290-297 describe the same results. Please avoid repetition.

Reviewer #2: This manuscript presents an extension of the Evolutionary Classification of Protein Domains (ECOD) to include AlphaFold Database (AFDB) structure predictions for all Swiss-Prot proteins. Using the Domain Parser for AlphaFold Models (DPAM) pipeline, the authors classify over 1 million domains derived from more than 540,000 protein structure predictions, achieving broad taxonomic and functional coverage. The study integrates structure-based domain classification with Swiss-Prot’s curated functional annotations and provides a comprehensive dataset deposited in Zenodo and the ECOD database.

Overall, this is a valuable resource because of the following strengths:

1. The work expands ECOD’s coverage into a major curated dataset (Swiss-Prot), representing an important addition to structural bioinformatics resources.

2. The authors describe a systematic and reproducible pipeline (DPAM + ECOD + Pfam integration), combining sequence, structure, and taxonomic analysis.

3. The reported high DPAM probability (mean 0.992) and extensive pLDDT validation underscore robust classification.

4. Data are openly available via Zenodo and ECOD.

5. The manuscript provides taxonomic, functional, and evolutionary insights—e.g., the discovery of >100,000 domains lacking Pfam mappings—highlighting areas for future functional annotation and structural validation.

However, there are still two issues for the authors to address:

• A more detailed analysis or discussion of the difference and similarity between ECOD extension with CATH’s use of AFDB can strengthen the work.

• Typographical errors (e.g., “Classificatioxn”) should be corrected.

**Have the authors made all data and (if applicable) computational code underlying the findings in their manuscript fully available?**

The PLOS Data policy requires authors to make all data and code underlying the findings described in their manuscript fully available without restriction, with rare exception (please refer to the Data Availability Statement in the manuscript PDF file). The data and code should be provided as part of the manuscript or its supporting information, or deposited to a public repository. For example, in addition to summary statistics, the data points behind means, medians and variance measures should be available. If there are restrictions on publicly sharing data or code —e.g. participant privacy or use of data from a third party—those must be specified.requires authors to make all data and code underlying the findings described in their manuscript fully available without restriction, with rare exception (please refer to the Data Availability Statement in the manuscript PDF file). The data and code should be provided as part of the manuscript or its supporting information, or deposited to a public repository. For example, in addition to summary statistics, the data points behind means, medians and variance measures should be available. If there are restrictions on publicly sharing data or code —e.g. participant privacy or use of data from a third party—those must be specified.requires authors to make all data and code underlying the findings described in their manuscript fully available without restriction, with rare exception (please refer to the Data Availability Statement in the manuscript PDF file). The data and code should be provided as part of the manuscript or its supporting information, or deposited to a public repository. For example, in addition to summary statistics, the data points behind means, medians and variance measures should be available. If there are restrictions on publicly sharing data or code —e.g. participant privacy or use of data from a third party—those must be specified.requires authors to make all data and code underlying the findings described in their manuscript fully available without restriction, with rare exception (please refer to the Data Availability Statement in the manuscript PDF file). The data and code should be provided as part of the manuscript or its supporting information, or deposited to a public repository. For example, in addition to summary statistics, the data points behind means, medians and variance measures should be available. If there are restrictions on publicly sharing data or code —e.g. participant privacy or use of data from a third party—those must be specified.

Reviewer #1: Yes

Reviewer #2: Yes

PLOS authors have the option to publish the peer review history of their article (what does this mean?). If published, this will include your full peer review and any attached files.). If published, this will include your full peer review and any attached files.). If published, this will include your full peer review and any attached files.). If published, this will include your full peer review and any attached files.

...

Reviewer #1: No

Reviewer #2: No

**Figure resubmission:**

**Reproducibility:**



---

## [Decision Letter · Decision Letter 1]

2 Feb 2026

PCOMPBIOL-D-25-01638R1

ECOD: Classification of domains in AFDB Swiss-Prot structure predictions

PLOS Computational Biology

Dear Dr. Schaeffer,

Thank you for submitting your manuscript to PLOS Computational Biology. After careful consideration, we feel that it has merit but does not fully meet PLOS Computational Biology's publication criteria as it currently stands. Therefore, we invite you to submit a revised version of the manuscript that addresses the points raised during the review process.

We look forward to receiving your revised manuscript.

Kind regards,

Chaok Seok

Academic Editor

PLOS Computational Biology

Nir Ben-Tal

Section Editor

PLOS Computational Biology

**Reviewers' comments:**

Reviewer's Responses to Questions

**Comments to the Authors:**

Reviewer #1: The revised version of the manuscript "ECOD: Classification of domains in AFDB Swiss-Prot structure predictions" by Grishin and co-workers essentially addressed all the concerns raised in the original submission. I recommend this manuscript for publication in PLOS Computational Biology after a couple of minor glitches are fixed.

1. In the responses, the authors stated that the advanced taxonomic search is available from the main ECOD page http://prodata.swmed.edu/ecod/af2_pdb/. It appears that this useful feature is available from http://prodata.swmed.edu/ecod/af2_pdb/search/advanced. Please make it available at the front page or provide an explicit link in the manuscript. It may also be beneficial to deploy an SSL connection to the server using Certbot or other services.

2. While the 4 superkingdom issue has been addressed in the revised version, line 93 still states "with domains spanning 4 superkingdoms, 98 phyla, and 10,254 species". Please fix.

Reviewer #3: In this manuscript, Schaeffer and co-workers apply the Domain Parser for AlphaFold

Models (DPAM) to annotate AlphaFold-predicted structures in the manually curated

UniProtKB/Swiss-Prot database. While the overall objective is timely and provides

valuable insight to related fields, several conceptual and presentation issues limit the

impact and interpretability of the work and should be revised for publication.

Major points:

1) A key result is the reported discrepancy between DPAM-parsed domains and Pfam

annotations. The authors attribute this largely to “novel evolutionary” relationships,

but this explanation is insufficiently justified. It remains unclear whether these

differences stem from structure-based versus sequence-based similarity,

differences in domain definition and annotation philosophy, or limitations of

AlphaFold predictions. The manuscript would benefit from a more rigorous

discussion supported by concrete examples.

2) Similar to pt 1), in lines 325-326, what does the fact that there are many

unannotated but with high-quality ordered structures mean? Are they truly new

domains, structures, that were absent in experimental DBs? Or is it due to disparity

in sequence similarity and structure similarity?

3) Although ECOD was published some time ago, still, a proper introduction of the

hierarchy, organization, and the rationale of the design the ECOD classification

framework should be provided. The meanings of the H-, T-, and X-level annotations

should be briefly explained to make the results accessible to a broader audience.

4) Even with rather low similarity, 40%, the number of singletons is almost the half of

the entire clusters. Does this mean that the current parsing scheme is too strict or

sensitive to small changes in structures? Authors should provide more explanation

and discussion on these large number of singletons.

5) Figure 2D is not referenced in the text and the color codes of Figure2D are not

presented in the legend.

6) Figure 3 is small and difficult to interpret.

Minor point:

1) In line 88, “Swiss-Prot/UniProtKB” should be corrected to “UniProtKB/Swiss-Prot.”

2) In line 313, pfam -> Pfam

**Have the authors made all data and (if applicable) computational code underlying the findings in their manuscript fully available?**

The PLOS Data policy requires authors to make all data and code underlying the findings described in their manuscript fully available without restriction, with rare exception (please refer to the Data Availability Statement in the manuscript PDF file). The data and code should be provided as part of the manuscript or its supporting information, or deposited to a public repository. For example, in addition to summary statistics, the data points behind means, medians and variance measures should be available. If there are restrictions on publicly sharing data or code —e.g. participant privacy or use of data from a third party—those must be specified.requires authors to make all data and code underlying the findings described in their manuscript fully available without restriction, with rare exception (please refer to the Data Availability Statement in the manuscript PDF file). The data and code should be provided as part of the manuscript or its supporting information, or deposited to a public repository. For example, in addition to summary statistics, the data points behind means, medians and variance measures should be available. If there are restrictions on publicly sharing data or code —e.g. participant privacy or use of data from a third party—those must be specified.requires authors to make all data and code underlying the findings described in their manuscript fully available without restriction, with rare exception (please refer to the Data Availability Statement in the manuscript PDF file). The data and code should be provided as part of the manuscript or its supporting information, or deposited to a public repository. For example, in addition to summary statistics, the data points behind means, medians and variance measures should be available. If there are restrictions on publicly sharing data or code —e.g. participant privacy or use of data from a third party—those must be specified.requires authors to make all data and code underlying the findings described in their manuscript fully available without restriction, with rare exception (please refer to the Data Availability Statement in the manuscript PDF file). The data and code should be provided as part of the manuscript or its supporting information, or deposited to a public repository. For example, in addition to summary statistics, the data points behind means, medians and variance measures should be available. If there are restrictions on publicly sharing data or code —e.g. participant privacy or use of data from a third party—those must be specified.

Reviewer #1: Yes

Reviewer #3: Yes

PLOS authors have the option to publish the peer review history of their article (what does this mean?). If published, this will include your full peer review and any attached files.). If published, this will include your full peer review and any attached files.). If published, this will include your full peer review and any attached files.). If published, this will include your full peer review and any attached files.

...

Reviewer #1: No

Reviewer #3: **Yes:** Juyong LeeJuyong LeeJuyong LeeJuyong Lee

**Figure resubmission:**
---

## [Decision Letter · Decision Letter 2]

17 Mar 2026

Dear Dr. Schaeffer,

We are pleased to inform you that your manuscript 'ECOD: Classification of domains in AFDB Swiss-Prot structure predictions' has been provisionally accepted for publication in PLOS Computational Biology.

Best regards,

Chaok Seok

Academic Editor

PLOS Computational Biology

Nir Ben-Tal

Section Editor

PLOS Computational Biology

Reviewer's Responses to Questions

**Comments to the Authors:**

Reviewer #1: The revised version of the manuscript has been further improved and addressed the majority of concerns raised in the previous submission. I recommend this manuscript for publication after a couple of minor issues are fixed

1) Font in the legend to Fig 4A is to small. Please fix

2) Individual panels in Fig 5 are not called within the main text

"At 99% identity, nearly all clusters contained 279 domains from a single phylum, consistent with recent evolutionary divergence (Fig. 5)" Expand the text to contextualize the data shown in Figs. 5A and 5B.

Reviewer #3: The authors addressed all my concerns successfully.

**Have the authors made all data and (if applicable) computational code underlying the findings in their manuscript fully available?**

The PLOS Data policy requires authors to make all data and code underlying the findings described in their manuscript fully available without restriction, with rare exception (please refer to the Data Availability Statement in the manuscript PDF file). The data and code should be provided as part of the manuscript or its supporting information, or deposited to a public repository. For example, in addition to summary statistics, the data points behind means, medians and variance measures should be available. If there are restrictions on publicly sharing data or code —e.g. participant privacy or use of data from a third party—those must be specified.requires authors to make all data and code underlying the findings described in their manuscript fully available without restriction, with rare exception (please refer to the Data Availability Statement in the manuscript PDF file). The data and code should be provided as part of the manuscript or its supporting information, or deposited to a public repository. For example, in addition to summary statistics, the data points behind means, medians and variance measures should be available. If there are restrictions on publicly sharing data or code —e.g. participant privacy or use of data from a third party—those must be specified.requires authors to make all data and code underlying the findings described in their manuscript fully available without restriction, with rare exception (please refer to the Data Availability Statement in the manuscript PDF file). The data and code should be provided as part of the manuscript or its supporting information, or deposited to a public repository. For example, in addition to summary statistics, the data points behind means, medians and variance measures should be available. If there are restrictions on publicly sharing data or code —e.g. participant privacy or use of data from a third party—those must be specified.requires authors to make all data and code underlying the findings described in their manuscript fully available without restriction, with rare exception (please refer to the Data Availability Statement in the manuscript PDF file). The data and code should be provided as part of the manuscript or its supporting information, or deposited to a public repository. For example, in addition to summary statistics, the data points behind means, medians and variance measures should be available. If there are restrictions on publicly sharing data or code —e.g. participant privacy or use of data from a third party—those must be specified.

Reviewer #1: None

Reviewer #3: Yes

PLOS authors have the option to publish the peer review history of their article (what does this mean?). If published, this will include your full peer review and any attached files.). If published, this will include your full peer review and any attached files.). If published, this will include your full peer review and any attached files.). If published, this will include your full peer review and any attached files.

...

Reviewer #1: No

Reviewer #3: **Yes:** Juyong LeeJuyong LeeJuyong LeeJuyong Lee

---

## [Editor Report · Acceptance letter]

PCOMPBIOL-D-25-01638R2

ECOD: Classification of domains in AFDB Swiss-Prot structure predictions

Dear Dr Schaeffer,

I am pleased to inform you that your manuscript has been formally accepted for publication in PLOS Computational Biology. Your manuscript is now with our production department and you will be notified of the publication date in due course.

With kind regards,

Anita Estes
